# Neonatal mortality in NHS maternity units by timing and mode of birth: a retrospective linked cohort study

Lucy Carty,[1] Christopher Grollman ![ORCID],[1] Rachel Plachcinski,[1] Mario Cortina-Borja,[2] Alison Macfarlane ![ORCID] [1]

¹Department of Midwifery, City, University of London, London, UK
²UCL Great Ormond Street Institute of Child Health, University College London, London, UK

**Correspondence to**
Professor Alison Macfarlane;
a.j.macfarlane@city.ac.uk

## ABSTRACT

**Objectives** To compare neonatal mortality in English hospitals by time of day and day of the week according to care pathway.

**Design** Retrospective cohort linking birth registration, birth notification and hospital episode data.

**Setting** National Health Service (NHS) hospitals in England.

**Participants** 6054536 liveborn singleton births from 2005 to 2014 in NHS maternity units in England.

**Main outcome measures** Neonatal mortality.

**Results** After adjustment for confounders, there was no significant difference in the odds of neonatal mortality attributed to asphyxia, anoxia or trauma outside of working hours compared with working hours for spontaneous births or instrumental births. Stratification of emergency caesareans by onset of labour showed no difference in mortality by birth timing for emergency caesareans with spontaneous or induced onset of labour. Higher odds of neonatal mortality attributed to asphyxia, anoxia or trauma out of hours for emergency caesareans without labour translated to a small absolute difference in mortality risk.

**Conclusions** The apparent 'weekend effect' may result from deaths among the relatively small numbers of babies who were coded as born by emergency caesarean section without labour outside normal working hours. Further research should investigate the potential contribution of care-seeking and community-based factors as well as the adequacy of staffing for managing these relatively unusual emergencies.

## INTRODUCTION

Since 2001, a considerable number of analyses have shown higher mortality rates among patients admitted to hospital at weekends compared with weekdays. There has been considerable debate about how to interpret these rates, with differences in quality of care and differences in case mix being frequently cited[1 2]; concerns about a 'weekend effect' led to a '7-day services' policy for the National Health Service (NHS) in England in 2015.[3] A systematic review looking at the need to increase specialist intensity at weekends for patients undergoing emergency hospital admission, published in 2021, concluded that

---

## STRENGTHS AND LIMITATIONS OF THIS STUDY

⇒ This analysis used a large linked data set bringing together data for over 6 million births over 10 years which enabled stratification by mode of onset of labour and mode of birth, and by time of day as well as day of birth.

⇒ Stillbirths: Stillbirths were excluded as around 90% of fetal deaths occur before the onset of labour so the timing of death is unknown.

⇒ Adjustment for covariates: The large number of variables available made it possible to adjust for covariates including obstetric risk factors and the seasonal and temporal nature of birth.

⇒ The subdivision of caesarean births by timing of decision into elective and emergency used in the UK has enabled the identification of a group which would not have been visible using the subdivision of caesareans into 'before labour' and 'in labour' used in many other countries.

⇒ Variables were all derived from anonymised hospital administrative data, so there were no data available about events in the community before admission to hospital or after discharge.

---

there was unlikely to be a single cause for the weekend effect. It pointed to the importance of case mix and concluded that the effect is unlikely to be an indicator of quality of hospital care.[4 5]

In the perinatal field, studies have investigated pregnancy outcomes by day of the week, time of the year and time of day, particularly from the 1970s onwards. Analyses of perinatal mortality in England and Wales among births in the years 1970–1976[6–8] and 1979–1996[9] showed higher mortality rates among babies born at weekends, but could only estimate crude rates, meaning no conclusions could be drawn. Studies which were able to adjust for confounding produced inconsistent results. A study in Canada found slightly higher crude rates of stillbirths and neonatal deaths among births at weekends, but the difference disappeared after adjustment for gestational age,[10] and a study in Australia found no difference

after adjustment for birth weight.[11] An analysis by day of the week in England published in 2015 concluded that perinatal mortality was highest at the weekend.[12] Some studies showed seasonal or short-term variation. Analyses of data for 1993–1995 from Wales[13] and Scotland,[14] with relatively small numbers of deaths, suggested a possible association of mortality with the rotation of junior doctors to new posts in August. An analysis for England and Wales for 1979–1996 showed seasonal variation, with higher perinatal mortality rates in winter than in summer.[9] Studies investigating time of day from Switzerland,[15 16] Sweden[17 18] and California[19] found higher mortality at night, although they reported varied risks in terms of type of perinatal death and time of night. An analysis of neonatal deaths at term in Scotland from 1985 to 2004 compared deaths among babies born from 09:00 to 17:00 on Mondays to Fridays to babies born outside these hours and days. It found higher rates of deaths ascribed to intra-partum anoxia among babies born 'out of hours' but no differences for other causes of death.[20]

The 2021 systematic review[4] recommended that further work should focus on underlying mechanisms and examine care processes in both hospital and the community. Such work has the potential to draw out the contributions of both case mix and staffing questions to observed higher mortality. In this analysis, we have analysed neonatal mortality of babies born during and outside of working hours in English hospitals according to mode of birth and onset of labour.

## METHODS
### Data
This study uses linked data from birth registration, birth notification and maternity hospital episode statistics.

Information about births in England and Wales is recorded in several systems. Socio-demographic information is recorded at birth registration. Further information, including gestational age and time of birth, is recorded at birth notification when each baby is issued with an NHS number. Information about care at birth in NHS hospitals in England is recorded in Maternity Hospital Episode Statistics (HES) within the mother-based HES deliveries file. Further information about the health of the baby and level of care required after birth is recorded in the baby-based HES baby file.

Following a series of pilot projects in collaboration with City, University of London, the Office for National Statistics (ONS) now routinely links birth registration and birth notification data. City, University of London has linked these data to HES and also to corresponding data for Wales to form the City Birth Cohort.[21] Authors had full access to the data from these previous efforts, and this study did not itself include further linkage.

### Selection of data for analysis
This study uses a source data set derived from the City Birth Cohort and consisting of all 6 054 536 singleton births occurring in NHS maternity units in England from 2005 to 2014 and with good links to HES. Derivation and analysis of linkage bias for this cohort has been described elsewhere.[22–24] In summary, it was possible to link over 94% of birth registration linked to notification records to HES delivery and birth records. The linkage rate increased over time as the quality of Maternity HES improved.

From this population, we removed live births occurring before 22 weeks of gestational age and births registered as stillborn. Nearly 90% of stillbirths in England are recorded as antepartum, with fetal death occurring before the onset of labour.[25] For the remaining stillbirths, we attempted to identify those where death occurred intrapartum but found that the timing of stillbirth was poorly recorded in both the Centre for Maternal and Child Enquiries (CMACE) data and the ONS birth registration data. As a result, for a substantial proportion of the relatively small number of records we could not determine whether the stillbirth was antepartum or intra-partum, so we chose to exclude all stillbirths from our outcome indicator.

By contrast, virtually all neonatal deaths are identified unambiguously so we have higher confidence in the completeness of that population. We used the ONS' modified Wigglesworth classification system[26] to classify neonatal deaths and remove deaths attributed to congenital anomalies. To derive the analysis population for modelling, we further removed births for which the time of birth was missing. We determined that recorded time of birth was subject to heaping on 5-min intervals but that there were no other important accumulation points.

### Statistical analysis
We used the mean and SD to summarise continuous variables and t-test for comparisons between groups. We used percentages to summarise categorical variables and $\chi^2$ test for comparisons between groups.

The primary outcome was cause-specific neonatal mortality attributed to asphyxia, anoxia or trauma using the ONS's modified Wigglesworth categories,[26 27] as this is the category of death most likely to be affected by quality of care and staffing factors relevant to the 'weekend effect'; deaths attributed to intrapartum anoxia were responsible for elevated out-of-hours mortality in a study of over 1 million singleton term live births in Scotland between 1985 and 2004.[20] We also modelled all-cause neonatal mortality unattributed to congenital anomaly (table 1).

We used an 8-day categorisation for day of birth which included Monday to Sunday and public holidays. Time of birth was categorised into daytime hours from 07:00 to 19:00 and night-time hours. A combination of day of birth and time of birth was used to classify births as occurring during 'working hours', defined as weekday daytime hours and 'out of hours', defined as all other times of the week and including all nights, weekends and holidays. In breaking down out-of-hours further, the hours

**Table 1** Neonatal deaths by modified Wigglesworth cause of death categories

| Causes of death | Neonatal deaths, n |
| --- | --- |
| **Cause arises before the onset of labour:** | |
| Congenital anomalies | 4070 |
| Antepartum infections | 489 |
| Immaturity-related conditions | 6178 |
| **Cause arises during, or shortly after labour and birth:** | |
| Asphyxia, anoxia or trauma | 1494 |
| **Cause arises after birth:** | |
| External conditions | 44 |
| Infections | 237 |
| Other specific conditions | 111 |
| Sudden infant deaths | 190 |
| **Unclassified:** | |
| Other conditions | 239 |
| Missing | 25 |
| **Total** | 13 077 |

from midnight to 07:00 on a Monday were classified as a weekend/holiday night to reflect the fact that people giving birth in that period would have had access to weekend staffing only in the preceding 48 hours.

The data set was stratified by mode of birth and later analysed by mode of onset of labour. To determine the mode of birth we used the Office of Population Censuses and Surveys (OPCS) procedure codes recorded in the standard HES record, supplemented by the HES 'maternity tail' variable DELMETH, as previously described.[22]

We used the HES maternity tail variable DELONSET to determine the mode of onset of labour. Models were constructed for each mode of birth with onset of labour as a covariate.

We identified candidate covariates and confounders fitting univariable and multivariable logistic regression models and used a combination of forwards and backwards selection to determine inclusion of covariates in the models. Models were compared using the likelihood ratio test and the Bayesian information criterion (BIC).[28]

Long-term trends, and mother's age were characterised using natural cubic splines with the amount of smoothing chosen by minimising BIC as a function of df. Day of birth in the year was modelled using yearly and semestral harmonic terms. Binary contrasts referred to sex of the baby and changes between Greenwich Mean Time and British Summer Time. Birth attendants notifying births are instructed to notify gestational age in completed weeks from last menstrual period but some may have used gestational age as recorded by ultrasound as this has become routine. Birth weight was measured in grams and categorised into five levels for modelling. Ethnicity as recorded at birth notification was coded using 17 categories based on the ethnicity question in the 2001 Census of England and Wales. Parity was defined as nulliparous or multiparous by combining information from ONS and HES and by reviewing linked HES records' Mother IDs to determine whether there had been previous births.[29]

We accounted for geographical variation by adjusting for the former Strategic Health Authority Region of England where the birth took place. Random effects terms were included in the models to allow for clustering of providers within the NHS trust where the birth took place. As the constituent hospitals making up some trusts varied over the 10-year time period, maternity units were allocated to a single trust (the 'Assigned Trust') for the entire period, even if they were not part of this Trust for the whole time, as described in the project report.[29] At the suggestion of a reviewer we checked whether the inclusion of trust could have adjusted away community effects that may have contributed to adverse outcomes. A sensitivity analysis excluding trust and using a generalised linear model without random effects showed no effect on the estimate for our birth timing variable. The risk of cause-specific and all-cause neonatal mortality was modelled fitting mixed-effects logistic regression models including a random effects term on the intercept to account for unobserved heterogeneity between Assigned Trusts.

During exploratory analyses we found associations between missing information for some variables and neonatal mortality. We also found non-random missingness patterns for key variables including gestational age and birth weight which were not suitable for imputation. For this reason, predictors were included in the models as categorical variables with a category for missing information. Data completeness, particularly for HES, has improved over time. We also observed this pattern in birth weight and gestational age data recorded at birth notification and registration; the percentage of births for which either birth weight or gestational age were missing was under 1%, well below the extent to which data were missing from HES. For around 15% of births the mode of onset of labour was unknown. They were included in the analyses that did not stratify by mode of onset, and we included them in models as an unknown category rather than considering them as missing data.

Following a suggestion from our patient and public involvement group, we calculated an absolute measure of risk—the number needed to harm (NNH)—as well as the relative measure presented by the OR. The NNH was calculated as:

$$NNH = ((PEER \times (OR - 1)) + 1) \div (PEER \times (OR - 1) \times (1 - PEER))$$

where the patient expected event rate (PEER) was the rate of neonatal death attributed to asphyxia, anoxia or trauma in the unexposed timing category and the OR was that from the adjusted logistic regression model.

Coding of all variables is shown in online supplemental table 1. All analyses were performed in the ONS Secure Research Service with R V.3.6.1.

## Patient and public involvement

Public involvement and engagement (PI&E) have been at the core of the work undertaken to develop and analyse the City Birth Cohort using a documented 'three-tier' approach.[30] Two service user representatives were involved in the earlier National Institute for Health Research-funded research from the outset as co-applicants. At the design stage, they contributed to the funding application and once funding was obtained they led the public engagement and involvement aspects of the project and helped to direct the focus of the research. In the Economic and Social Research Council-funded research described here, service representatives from Maternity Voices Partnerships (MVPs) became involved. The PI&E strategy was based on the same three-tier model.[30] For this particular project our PI&E lead presented previous findings and our new plans at a national event for MVP lay members in order to collect the views of a wider constituency of parents, and also invited expressions of interest in joining a new PI&E advisory group. Three people (white British, black British and Asian British) joined the group. Another MVP member joined our external advisory group at a later date. The PI&E group has met with the PI&E lead and data analyst regularly throughout the project to discuss progress and findings, including moving online during the COVID-19 pandemic. They have raised issues, such as the level of detail available on ethnicity, which have fed into the models, and stressed the importance of contextualising messages for both lay and clinical audiences. Group members have worked with the PI&E lead and data analyst to develop guides to publicly available data sets and reports on five topics of particular concern to them: ethnicity in maternity statistics, miscarriage, birth interventions, prematurity and smoking in pregnancy. They also presented their learning, experiences and recommendations for future data intensive research at a dissemination event for parents and service user representatives. A second, multidisciplinary, event is planned in spring 2023 and we will also share our results at future MVP conferences.

## RESULTS

### Numbers and causes of deaths in the population

The source data set consisted of 6 054 536 singleton births taking place in NHS maternity units in England between 1 January 2005 and 31 December 2014. We excluded 25 748 births recorded as stillborn (4.25 per 1000 births) and 1782 births recorded as occurring at less than 22 weeks gestational age (0.03%).

Table 1 shows neonatal deaths of babies born alive after 22 weeks of gestation. The group of intrapartum causes—asphyxia, anoxia and trauma—caused the third highest number of neonatal deaths. The neonatal mortality rate for deaths attributed to asphyxia, anoxia or trauma was 0.25 per 1000 live births.

We further excluded 6494 births recorded as infant deaths (including 4070 neonatal deaths) attributed to

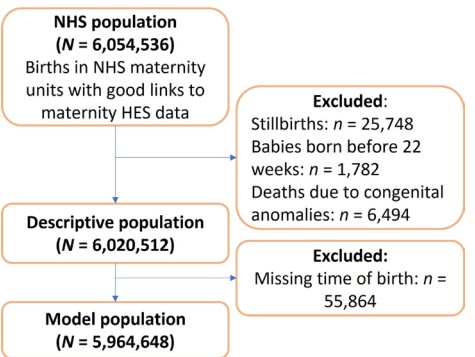

**Figure 1** Population. HES, Hospital Episode Statistics; NHS, National Health Service.

congenital anomalies (0.11%), leaving a descriptive population of 6 020 512 births (figure 1).

### Characteristics of population for analysis of timing of birth

To derive the analysis population, we then excluded 55 864 births that had missing information about time of birth as we could not categorise these as exposed or unexposed. We did not find any evidence that missing time of birth was associated with either neonatal mortality ($\chi^2$ p=0.75) or neonatal mortality attributed to asphyxia, anoxia or trauma ($\chi^2$ p=1).

Baseline characteristics in the analysis population (N=5 964 648) were stratified by whether births took place inside weekday working hours (weekdays 07:00 to 19:00) or outside working hours (nights, weekends and holidays) (table 2). The proportion of births that took place outside of weekday working hours during nights, weekends or holidays was 61.0%, lower than the 65.1% of hours that were outside working hours. The standardised mean difference was small for all characteristics (p<0.10), except for mother's age, parity and previous caesareans. Among multiparous women, 32.0% of births during working hours were to people who had previously had a caesarean compared with 10.5% of those out of hours. Birth during working hours also included higher proportions of older mothers.

### Crude neonatal mortality rates

Crude rates of cause-specific neonatal mortality attributed to asphyxia, anoxia or trauma were higher for births out of hours than for births during working hours (0.27 compared with 0.21 per 1000 live births, rate ratio 1.26, 95% CI 1.13 to 1.40). The crude mortality rate ratio was not significantly different from 1 for spontaneous, instrumental or emergency caesarean births. The elevated rate in the overall population was driven by a large rate ratio among births recorded as planned caesareans but occurring out of hours (table 3), an estimate which is likely to be highly biased due to misclassification, as we discuss below. Due to this issue of misclassification, we did not include planned caesareans in the modelling analyses. Results were similar for all-cause mortality, except that the crude mortality rate ratio was significantly below 1 for emergency caesarean births (0.86, 95% CI 0.80 to 0.93),

**Table 2** Characteristics of births in analysis population inside and outside of weekday working hours

| Characteristic | Level | Working hours | % | Out of hours | % | P value | SMD |
|---|---|---|---|---|---|---|---|
| N | | 2 328 731 | | 3 635 917 | | | |
| Baby sex | Female | 1 134 931 | 48.7 | 1 766 765 | 48.6 | 0.001 | 0.003 |
| | Male | 1 193 800 | 51.3 | 1 869 152 | 51.4 | | |
| Gestational age | 22–28 weeks | 6764 | 0.3 | 12 025 | 0.3 | <0.001 | 0.063 |
| | 28–32 weeks | 15 219 | 0.7 | 21 832 | 0.6 | | |
| | 32–37 weeks | 115 228 | 4.9 | 175 908 | 4.8 | | |
| | 37–42 weeks | 2 097 955 | 90.1 | 3 236 055 | 89.0 | | |
| | 42 weeks and over | 76 852 | 3.3 | 163 639 | 4.5 | | |
| | Missing | 16 713 | 0.7 | 26 458 | 0.7 | | |
| Birth weight | Under 2500 | 129 603 | 5.6 | 204 111 | 5.6 | <0.001 | 0.012 |
| | 2500–2999 | 369 786 | 15.9 | 589 999 | 16.2 | | |
| | 3000–3499 | 848 644 | 36.4 | 1 318 645 | 36.3 | | |
| | 3500–3999 | 694 708 | 29.8 | 1 086 195 | 29.9 | | |
| | 4000 and over | 272 305 | 11.7 | 415 955 | 11.4 | | |
| | Missing or unfeasible | 13 685 | 0.6 | 21 012 | 0.6 | | |
| Mother's age | Under 20 | 115 846 | 5.0 | 220 044 | 6.1 | <0.001 | 0.116 |
| | 20–24 | 401 146 | 17.2 | 711 565 | 19.6 | | |
| | 25–29 | 618 854 | 26.6 | 1 019 383 | 28.0 | | |
| | 30–34 | 684 650 | 29.4 | 1 025 162 | 28.2 | | |
| | 35–39 | 407 151 | 17.5 | 539 755 | 14.8 | | |
| | Over 40 | 101 084 | 4.3 | 120 008 | 3.3 | | |
| Parity | Nulliparous | 892 405 | 38.3 | 1 594 567 | 43.9 | <0.001 | 0.114 |
| | Moderate: 1–4 | 1 367 620 | 58.7 | 1 935 522 | 53.2 | | |
| | High: 5–9 | 66 080 | 2.8 | 101 595 | 2.8 | | |
| | Very high: 10–14 | 2409 | 0.1 | 3905 | 0.1 | | |
| | Extreme/unfeasible | 169 | 0.0 | 257 | 0.0 | | |
| | Missing | 48 | 0.0 | 71 | 0.0 | | |
| Ethnicity | White British | 1 491 685 | 64.1 | 2 319 066 | 63.8 | <0.001 | 0.019 |
| | White Irish | 14 809 | 0.6 | 22 483 | 0.6 | | |
| | Any other white background | 175 116 | 7.5 | 274 915 | 7.6 | | |
| | Mixed white and black Caribbean | 23 427 | 1.0 | 38 851 | 1.1 | | |
| | Mixed white and black African | 14 880 | 0.6 | 23 188 | 0.6 | | |
| | Mixed white and Asian | 24 680 | 1.1 | 38 546 | 1.1 | | |
| | Any other mixed background | 41 207 | 1.8 | 64 505 | 1.8 | | |
| | Asian Indian | 71 267 | 3.1 | 111 751 | 3.1 | | |
| | Asian Pakistani | 94 864 | 4.1 | 155 558 | 4.3 | | |
| | Asian Bangladeshi | 32 892 | 1.4 | 53 091 | 1.5 | | |
| | Any other Asian background | 37 163 | 1.6 | 56 125 | 1.5 | | |
| | Black Caribbean | 21 566 | 0.9 | 35 681 | 1.0 | | |
| | Black African | 74 878 | 3.2 | 111 342 | 3.1 | | |

**Table 2** Continued

| Characteristic | Level | Working hours | % | Out of hours | % | P value | SMD |
|---|---|---|---|---|---|---|---|
| | Any other black background | 17003 | 0.7 | 26458 | 0.7 | | |
| | Chinese | 11253 | 0.5 | 18536 | 0.5 | | |
| | Any other ethnic group | 48855 | 2.1 | 74154 | 2.0 | | |
| | Not known | 133186 | 5.7 | 211667 | 5.8 | | |
| Region | North East | 113142 | 4.9 | 173066 | 4.8 | <0.001 | 0.022 |
| | North West | 314194 | 13.5 | 490345 | 13.5 | | |
| | Yorkshire/Humber | 228995 | 9.8 | 373778 | 10.3 | | |
| | East Midlands | 173054 | 7.4 | 273713 | 7.5 | | |
| | West Midlands | 254354 | 10.9 | 405784 | 11.2 | | |
| | East of England | 231783 | 10.0 | 358103 | 9.8 | | |
| | London | 461053 | 19.8 | 704290 | 19.4 | | |
| | South East Coast | 186082 | 8.0 | 282432 | 7.8 | | |
| | South West | 188108 | 8.1 | 299581 | 8.2 | | |
| | South Central | 177966 | 7.6 | 274825 | 7.6 | | |
| Marital status at birth registration | Joint registration different address | 215147 | 9.2 | 366533 | 10.1 | <0.001 | 0.054 |
| | Joint registration same address | 681512 | 29.3 | 1111692 | 30.6 | | |
| | Sole registration | 134867 | 5.8 | 227363 | 6.3 | | |
| | Within marriage | 1297198 | 55.7 | 1930320 | 53.1 | | |
| | Missing | 7 | 0.0 | 9 | 0.0 | | |
| Mother's country of birth | UK | 1743087 | 74.9 | 2729074 | 75.1 | <0.001 | 0.016 |
| | Other Europe | 185255 | 8.0 | 296498 | 8.2 | | |
| | Africa | 128846 | 5.5 | 190826 | 5.2 | | |
| | The Americas and the Caribbean | 36325 | 1.6 | 53466 | 1.5 | | |
| | Middle East and Asia | 221835 | 9.5 | 346122 | 9.5 | | |
| | Oceania and Antarctica | 10477 | 0.5 | 15475 | 0.4 | | |
| | Missing | 1453 | 0.1 | 2228 | 0.1 | | |
| Previous caesarean birth (multiparous women only) | No | 976973 | 68.0 | 1826456 | 89.5 | <0.001 | 0.544 |
| | Yes | 459188 | 32.0 | 214645 | 10.5 | | |

SMD, standardised mean difference.

meaning a lower crude all-cause neonatal mortality rate for births out of hours than for births during working hours (online supplemental table 2).

## Adjusted odds of neonatal mortality by type of birth

We fitted multivariable mixed-effects logistic regression models to isolate the effect of working hours from potential confounding by other factors associated with neonatal mortality attributed to asphyxia, anoxia or trauma. Our final models were adjusted for baby's sex, gestational age, birth weight, mode of onset of labour, geographical region, NHS trust, baby's ethnicity, baby's date of birth, mother's age and parity, as well as yearly harmonic terms for day of year of the birth. After adjustment, there was no significant difference in the odds of mortality out of hours compared with working hours for spontaneous or instrumental births. The odds of neonatal mortality attributed to asphyxia, anoxia or trauma for emergency caesareans was significantly higher out of hours compared with

**Table 3** Crude cause-specific neonatal mortality attributed to asphyxia, anoxia or trauma by time and day of birth, stratified by mode of birth

|  | Neonatal deaths | Total live births | Rate* | Rate ratio | 95% CI |
|---|---|---|---|---|---|
| **Total population** |  |  |  |  |  |
| Working hours | 499 | 2 328 731 | 0.21 |  |  |
| Out of hours | 981 | 3 635 917 | 0.27 | 1.26 | 1.13 to 1.40 |
| **Spontaneous birth** |  |  |  |  |  |
| Working hours | 115 | 1 202 710 | 0.10 |  |  |
| Out of hours | 264 | 2 558 357 | 0.10 | 1.08 | 0.87 to 1.35 |
| **Instrumental birth** |  |  |  |  |  |
| Working hours | 60 | 256 452 | 0.23 |  |  |
| Out of hours | 107 | 485 425 | 0.22 | 0.94 | 0.69 to 1.30 |
| **Emergency caesarean** |  |  |  |  |  |
| Working hours | 301 | 311 218 | 0.97 |  |  |
| Out of hours | 587 | 554 974 | 1.06 | 1.09 | 0.95 to 1.26 |
| **Planned caesarean** |  |  |  |  |  |
| Working hours | 21 | 553 117 | 0.04 |  |  |
| Out of hours | 16 | 29 873 | 0.54 | 14.11 | 7.25 to 26.94 |

*Per 1000 live births.

during working hours (OR, 1.21; 95% CI 1.05 to 1.39) (table 4). For all-cause neonatal mortality, the ORs were not significantly different from 1 for any mode of birth (online supplemental table 3).

### Onset of labour for emergency caesarean births

We further stratified emergency caesareans by mode of onset of labour. The largest group of emergency caesareans occurred after spontaneous onset, followed by induced onset, and fewer than one-fifth are emergency caesareans before the start of labour. These emergency caesarean births without labour constituted 2.2% of all births in the data set. The rates of neonatal mortality were highest among the group with no labour, and that is also the only group for which the rate of neonatal mortality was significantly higher out of hours compared with during working hours (table 5).

The odds of neonatal mortality attributed to asphyxia, anoxia or trauma for emergency caesareans without labour were two-thirds higher for births out of hours compared with births during working hours, unaffected by adjustment for characteristics of the mother, baby and birth (OR, 1.66; 95% CI 1.28 to 2.16) (table 6). No significant difference by working hours was seen in neonatal mortality attributed to asphyxia, anoxia or trauma for emergency caesareans with spontaneous or induced onset.

Babies born by emergency caesarean without labour are a group at inherently high risk, so to estimate the effect of out-of-hours care we further adjusted the model for this group of births for obstetrical risk factors. We compared obstetrical risk factor characteristics during and out of working hours (online supplemental table 4) and included in the model those that improved fit. After adjustment, there remained 48% higher odds of neonatal mortality attributed to asphyxia, anoxia or trauma out of hours compared with working hours (OR 1.48; 95% CI

**Table 4** Unadjusted and adjusted ORs for neonatal mortality attributed to asphyxia, anoxia or trauma out of hours compared with during working hours, by type of birth

|  | Out of hours compared with working hours | | | |
|---|---|---|---|---|
|  | Unadjusted OR | 95% CI | Adjusted OR* | 95% CI |
| **Neonatal mortality attributed to asphyxia, anoxia or trauma** |  |  |  |  |
| Spontaneous birth | 1.08 | 0.87 to 1.34 | 1.09 | 0.87 to 1.35 |
| Instrumental birth | 0.94 | 0.69 to 1.29 | 0.96 | 0.70 to 1.32 |
| Emergency caesarean | 1.09 | 0.95 to 1.26 | 1.21 | 1.05 to 1.39 |

*Adjusted for baby's sex, gestational age, birth weight, mode of onset of labour, geographical region, NHS trust, baby's ethnicity, baby's date of birth, maternal age and maternal parity, as well as yearly harmonic terms for day of year of the birth and a natural spline for day of the study period.

**Table 5** Crude cause-specific neonatal mortality attributed to asphyxia, anoxia or trauma by time and day of birth, among emergency caesareans births, stratified by mode of onset of labour

| | Neonatal deaths | Total live births | Rate* | Rate ratio | 95% CI |
|---|---|---|---|---|---|
| **Spontaneous onset** | | | | | |
| Working hours | 103 | 120 791 | 0.85 | | |
| Out of hours | 204 | 226 528 | 0.90 | 1.06 | 0.84 to 1.34 |
| **Induced onset** | | | | | |
| Working hours | 29 | 75 513 | 0.38 | | |
| Out of hours | 71 | 162 353 | 0.44 | 1.14 | 0.75 to 1.78 |
| **No labour** | | | | | |
| Working hours | 86 | 60 202 | 1.43 | | |
| Out of hours | 169 | 71 700 | 2.36 | 1.65 | 1.28 to 2.15 |

*Per 1000 live births.

1.14 to 1.92) (table 6). The full coefficients in the fully adjusted model are presented in online supplemental table 5.

### Timing out of hours

To understand better the circumstances in which births were at higher risk of neonatal mortality we disaggregated the 'out of hours' category to allow us to distinguish evenings from weekends and holidays. Among emergency caesarean births without labour, being born during daytime hours of 07:00 to 19:00 at the weekend or on a holiday held no greater risk of mortality attributed to asphyxia, anoxia or trauma than being born during daytime hours on a working weekday. By contrast, emergency caesarean birth without labour in the night-time held an increased risk both during weeknights (OR=1.56, 95% CI 1.15 to 2.11) and night-time at weekends and holidays (OR=1.75, 95% CI 1.24 to 2.47) (table 7). There still was no association between mortality risk and working hours for spontaneous or instrumental birth or emergency caesareans with spontaneous/induced onset using the four-category timing variable (data not shown).

We further investigated whether risk of death out of hours was significantly associated with gestational age, local-area deprivation (using deciles of the Index of Multiple Deprivation) or year of birth (data not shown). We found that the only gestational age category where birth out of hours had a significant crude rate ratio for death attributed to asphyxia, anoxia or trauma compared with working hours was among term births (37–42 weeks). We found no evidence that the association of birth timing with mortality differed by epoch of birth (characterised in 2-year epochs and in 5-year epochs across the period 2005–2014). Lower local-area deprivation appeared to have had a small protective effect but this was not significant.

### Absolute risks from emergency caesarean without labour out of hours

We calculated how many emergency caesareans without labour would have to occur out of hours to be associated with a neonatal death attributed to asphyxia, anoxia or trauma beyond those expected if all emergency caesareans without labour happened during working hours. The PEER was taken as the rate of neonatal death attributed to asphyxia, anoxia or trauma among emergency caesareans without labour during weekday working hours, 1.43 per 1000 live births (table 5), and the ORs from the model further adjusted for obstetrical risk factors were 1.56 for weekday night-time and 1.75

**Table 6** Unadjusted and adjusted ORs for neonatal mortality attributed to asphyxia, anoxia or trauma out of hours compared with during working hours for emergency caesareans

| | ORs, out of hours compared with during working hours | | | | | |
|---|---|---|---|---|---|---|
| Mode of onset | Unadjusted | 95% CI | Adjusted for mother, baby and birth characteristics* | 95% CI | Further adjusted for obstetrical risk factors† | 95% CI |
| **Spontaneous** | 1.06 | 0.83 to 1.34 | 1.06 | 0.83 to 1.34 | — | — |
| **Induced** | 1.14 | 0.74 to 1.75 | 1.14 | 0.74 to 1.76 | — | — |
| **Caesarean (no labour)** | 1.65 | 1.27 to 2.14 | 1.66 | 1.28 to 2.16 | 1.48 | 1.14 to 1.92 |

*Adjusted for baby's sex, gestational age, birth weight, geographical region, NHS trust, baby's ethnicity, baby's date of birth, maternal age and maternal parity, as well as yearly harmonic terms for day of year of the birth.
†Further adjusted for placental abruption, maternal care for abnormality of pelvic organs, malpresentation of the fetus, pre-eclampsia, postpartum haemorrhage and antepartum haemorrhage.

**Table 7** Crude numbers and rates, and modelled ORs, for neonatal mortality attributed to asphyxia, anoxia or trauma out of hours compared with weekday daytime working hours for emergency caesareans without labour

|  | Neonatal deaths | Total live births | Crude rate* | Adjusted† OR | 95% CI |
|---|---|---|---|---|---|
| **Weekday daytime** | 86 | 60 202 | 1.43 | *Ref* | *Ref* |
| **Weekday night-time** | 86 | 33 558 | 2.56 | 1.56 | 1.15 to 2.11 |
| **Weekend/holiday daytime** | 29 | 20 068 | 1.45 | 0.95 | 0.63 to 1.46 |
| **Weekend/holiday night-time** | 54 | 18 074 | 2.99 | 1.75 | 1.24 to 2.47 |

*Per 1000 live births.
†Adjusted for baby's sex, gestational age, birth weight, geographical region, NHS trust, baby's ethnicity, baby's date of birth, maternal age and maternal parity, as well as yearly harmonic terms for day of year of the birth and presence of obstetrical risk factors (placental abruption, maternal care for abnormality of pelvic organs, malpresentation of the fetus, pre-eclampsia, postpartum haemorrhage and antepartum haemorrhage).

for weekend/holiday night-time (table 7). The resulting NNHs were 1258 for weekday nights and 933 for weekend/holiday nights. This means that for every 1258 births by emergency caesarean without labour born on a weekday night, there would be one additional neonatal death attributed to asphyxia, anoxia or trauma above what would have happened if all those births had been in weekday working hours; for such births on a weekend/holiday night, there would be one additional neonatal death attributed to asphyxia, anoxia or trauma in every 933 births. This amounts to 46 additional neonatal deaths in England across the 10-year study period, or between three and six deaths per year (online supplemental table 6). It constitutes 18% of the 255 neonatal deaths attributed to asphyxia, anoxia or trauma occurring among emergency caesareans without labour, 3% of the 1494 neonatal deaths attributed to asphyxia, anoxia or trauma and 0.4% of all neonatal deaths.

## DISCUSSION
### Statement of principal findings
Overall, this study did not find evidence of a higher risk of all-cause or cause-specific neonatal mortality for spontaneous or instrumental births, or for births by emergency caesarean after spontaneous or induced onset, when born outside of working hours.

Among babies coded as born by emergency caesarean without labour, which accounted for 2% of births in our analysis population, those born at night-time during the week had 56% higher odds of neonatal death attributed to asphyxia, anoxia or trauma when compared with those born during weekday working hours (defined as between 07:00 and 19:00 Monday to Friday); those born at night-time during the weekend or on a holiday had 75% higher odds compared with those born in weekday working hours. In our data set, with 71 700 births coded as emergency caesareans without labour outside working hours, we estimate that 46 excess deaths are likely to be associated with being born outside of working hours across the 10 years.[31]

### Strengths and weaknesses of the study and strengths and weaknesses in relation to other studies, discussing important differences in results
#### Strengths related to the data source
This study used a large, linked data set which brings together information on birth registration, birth notification and hospital maternity data for over 6 million births over 10 years. To date, a cohort of this size has not been used to analyse birth outcomes based on timing in the NHS. The large cohort allowed for stratification by distinct care pathways based on both onset of labour and mode of birth, and for the opportunity to examine mortality attributed to asphyxia, anoxia or trauma, most likely to be affected by care at birth but comprising approximately 11% of neonatal deaths. A study in Scotland found a raised risk of mortality attributed to anoxia in term births[20] but because of the smaller population, the data related to 1 039 560 live births over the 20 years 1985–2004 and the numbers were not large enough to stratify by onset of labour and type of birth. The study used data from the Scottish Stillbirth and Infant Death Survey in which information from death certificates was supplemented by further information recorded in clinical settings and classified using a modification of the Wigglesworth classification. The definition of anoxia was 'broad, including hypoxia, acidosis and asphyxia'.[20] The ONS modified Wigglesworth classification[26] had been designed to classify conditions recorded on stillbirth and neonatal death certificates and does not use data from other sources. It groups together anoxia, asphyxia and trauma and so the inclusion criteria in our study do not differ significantly from those used in the earlier study.

Moreover, the analysis of Scottish data included only term births whereas our large data set allowed for adjustment by gestational age. Our findings confirm those of this Scottish report but further identify a specific cohort of births at risk: the subset of births born by emergency caesarean without labour.

The nations of the UK are unusual in stratifying caesarean sections into elective/planned and emergency, while in many other countries the subdivision is into whether they take place before or in labour.[32] The

subgroup we have identified would not have been visible if the latter categorisation had been used. An audit of caesarean section in Scotland in 1994/1995 found, at a time when the overall caesarean section rate was lower, that 14.1% of caesarean sections were classified as emergency sections before the onset of labour and that these were mainly attributed to fetal growth restriction or fetal distress.[33]

Given the large number of variables available in the linked data sets, we were able to adjust for relevant covariates including obstetrical risk factors, unlike the Scottish study. We have also adjusted for the seasonal and temporal nature of births and neonatal mortality over the study period, reported in our previous analyses of this data set.[22]

### Strengths and limitations related to fetal death classification

A study using data from Maternity HES to analyse perinatal mortality by day of the week concluded that mortality was higher at weekends.[34] Without linkage to birth notification, it was unable to take account of the time of day and the authors' analysis and interpretation of their results was highly criticised in subsequent rapid responses.[35–37] Like past analyses of rates of stillbirth and perinatal mortality in England and Wales, this analysis made no distinction between intrapartum and antepartum stillbirths, despite the fact that the majority of stillbirths are antepartum and are therefore most are unlikely to be affected by care at birth. In contrast to other studies, we removed stillbirths from our analyses. To investigate intrapartum stillbirths poses the problem of identifying which stillbirths were definitely intrapartum when for many stillbirths it is unclear whether they occurred before or during labour. One of our advisors, the late Martin Ward Platt, suggested that we obtain confidential enquiry data to supplement the information from stillbirth registration certificates, following the precedent of the Scottish Stillbirth and Infant Death Survey, but it proved impossible to obtain a consistent series of data because of the many changes which took place in the confidential enquiries over the years 2005–2014. Permission was obtained from the Healthcare Quality Improvement Partnership to access data compiled by Confidential Enquiry into Maternal and Child Health (CEMACH) and CMACE for births in the years 2005–2010, in which there were a number of year-to-year changes.[21] More substantial changes were made from 2012 when responsibility for the confidential enquiry programme passed to Mothers and Babies: Reducing Risk through Audits and Confidential Enquiries across the UK (MBRRACE-UK). Its most recent report found that of the 1939 stillbirths in England in 2020, 89% were antepartum. Of the remainder 141 were classified as occurring intrapartum and 79 being of unknown timing.[38] In view of this continuing difficulty, we did not continue our plans to analyse intrapartum stillbirths.

### Strengths and limitations of classification of caesareans

Our classification of caesareans requires decision-making based on how births are coded, which may not always reflect clinical practice and, particularly for planned caesareans, may be influenced by clinical decisions made at earlier 'booking' antenatal appointments. For births recorded as planned caesareans, we found evidence of an inflated rate ratio for all-cause and cause-specific mortality outside of working hours which may be related to data recording and requires further consideration. Only 29 873 of the 582 990 planned caesareans in our analysis population (5%) took place outside of working hours. Among 'planned caesareans', the underlying population differed according to working hours: planned caesareans usually occur in working hours, and those 'planned caesareans' that occur out of hours are likely to involve complications. An audit of caesarean section in Scotland in 1994/1995 found that elective caesareans outside the hours of 09:00 to 18:00 occurred mainly where women were booked for an elective caesarean but went into labour before the planned date.[33] A comparison of OPCS procedure codes and the DELMETH variable from the HES maternity tail published in 2013 found that mode of delivery being recorded as emergency caesarean in one source and planned caesarean in the other was by far the most common type of inconsistency.[31] The scheduling of elective repeat caesareans during working hours is likely to explain the association of previous caesarean birth with birth during working hours as theses have previously been shown to take place primarily on weekday mornings 09:00 to 11:00.[22] Similarly, higher caesarean rates among older women could explain the higher proportions of births during working hours among older age groups.

We suspect that the crude findings related to planned caesareans were due to misclassification of actual emergency caesareans as 'planned', but there is little way to verify this without booking information and this requires further investigation by other means. There could be an important subgroup of women who are identified as at risk at booking but then progress to become emergencies. The limitations of the administrative data in this study meant we were unable to assess whether this is related to any differences in care in and out of hours.

### Additional strengths and limitations

Studies have used 09:00 to 17:00 to represent working hours, but are unlikely to reflect typical working hours in the NHS. We used a wider definition of working hours 07:00 to 19:00 to capture this and to reduce the potential for misclassification.

A key strength is our patient engagement—we have presented the number needed to harm through recommendations from maternity service users who felt that ORs and relative risks were not accessible for all to understand and did not fully capture rarity of events and absolute risk. This is also important for planning and strategies. To our knowledge, this is the first time that the

NNH has been estimated for neonatal mortality in relation to hospital working hours and birth type.

As with all observational studies, there is a potential for unmeasured confounding despite adjustment for multiple covariates. There is also potential for variation in reporting which can vary between trusts and over time, although the quality and completeness of HES improved over the period covered by this study. Future analyses are needed to confirm if these results persisted after the study period as the introduction in 2015 of a second national system, the Maternity Services Data set may have competed for staff time available for data recording. Investigations of the 'weekend effect' assume that timing of birth is associated with availability of specialist care, but we did not have any data to directly measure that availability. The size and completeness of our data set gives us confidence in the accuracy of our findings for NHS births in England as a whole, although variation within England will not be captured in this estimate.

### Meaning of the study: possible explanations and implications for clinicians and policymakers

Overall, out of hours care is not associated with a raised risk of neonatal mortality. Policy should focus on making arrangements for the small subset of emergencies where there may be an association rather than regarding all births out of hours as dangerous. Such arrangements may regard antenatal monitoring of or advice on healthcare-seeking behaviour for parents of particularly vulnerable babies.

Our NNH finds that over a 10-year period there were approximately 46 excess neonatal deaths attributed to asphyxia, anoxia or trauma among babies born out of hours by emergency caesarean without labour.

Emergency caesareans without labour make up approximately 2% of all births over the study period. Compared with other care pathways, they have by far the highest crude rates of neonatal mortality both overall and of mortality attributed to asphyxia, anoxia or trauma, both in and out of working hours.

As such, these births represent a cohort of mothers and babies with high risk, emergent conditions that are, by nature, likely to be unpredictable. The 2017 MBRRACE report on term, singleton, intrapartum stillbirth and intrapartum-related neonatal deaths reported that many of the deaths they reviewed followed pregnancies classified as low-risk.[39] That report also found that in at least a quarter of the 78 cases reviewed, sampled randomly from all eligible deaths, staffing capacity issues played a role in the death. A systematic review of articles on weekend mortality did not establish clear reasons for the higher weekend mortality for general NHS A&E care.[4] That review found no weekend effect in maternity admissions[40] although these were not disaggregated as we have done, and the weekday/weekend difference may elicit different findings that the working/out of hours classification that we used. That study also did not include comparisons of daytime and night-time, which our results suggest may be more important than the weekday/weekend distinction for the small group of births where working hours is associated with higher risk. The present study presents a comparable pattern of increased risk for births that reflect 'A&E-like' emergency maternal care, women not in labour but with conditions which indicate an immediate caesarean. Possible factors of influence might include differences in community healthcare or healthcare-seeking behaviour out of hours, availability of staff out of hours to take calls querying symptoms or to provide procedures such as scans and transport to maternity units being more difficult out of hours. This could lead to differences in case presentation with maternity users presenting with conditions that require emergency caesareans later than during working hours.

### Unanswered questions and future research

We have found that women facing an emergency requiring caesarean birth without labour have a small increased risk of neonatal mortality when giving birth at night. However, the reasons behind this remain unclear, particularly whether the increased risk has its origins in hospital or community care.

Future research should focus on characterising the cohort of women who have an emergency caesarean without labour, whether there are aspects of their care pathways or care-seeking behaviour that would be amenable to intervention and whether a planned caesarean would have been an option.

Analyses by NHS trust would not include sufficient numbers of deaths to look at mortality outcomes but could provide useful information on rates of emergency caesareans in and out of hours and ideally could link to information on staff-to-patient ratios. We had hoped to do this in our earlier project using NHS workforce statistics, but found that they were too inconsistent.[21] There may be scope to use the City Birth Cohort for more fine-grained investigation including of variation by trust, if power could be increased using a neonatal near-miss outcome rather than mortality.[41]

### Licence for publication

The Corresponding Author has the right to grant on behalf of all authors and does grant on behalf of all authors, a worldwide licence to the Publishers and its licensees in perpetuity, in all forms, formats and media (whether known now or created in the future), to (1) publish, reproduce, distribute, display and store the Contribution, (2) translate the Contribution into other languages, create adaptations, reprints, include within collections and create summaries, extracts and/or, abstracts of the Contribution, (3) create any other derivative work(s) based on the Contribution, (4) to exploit all subsidiary rights in the Contribution, (5) the inclusion of electronic links from the Contribution to third party material wherever it may be located; and, (6) licence any third party to do any or all of the above.

The Submitting Author has the right to grant and does grant on behalf of all authors of the Work an exclusive licence on a worldwide, perpetual, irrevocable, royalty-free basis to BMJ Publishing Group Ltd ('BMJ') and its licensees.

The Submitting Author accepts and understands that any supply made under these terms is made by BMJ to the Submitting Author unless you are acting as an employee on behalf of your employer or a postgraduate student of an affiliated institution which is paying any applicable article publishing charge ('APC') for Open Access articles. Where the Submitting Author wishes to make the Work available on an Open Access basis (and intends to pay the relevant APC), the terms of reuse of such Open Access shall be governed by a Creative Commons licence—details of these licences and which licence will apply to this Work are set out in our licence referred to above.

## TRANSPARENCY STATEMENT

The manuscript's guarantor affirms that this manuscript is an honest, accurate and transparent account of the study being reported; that no important aspects of the study have been omitted; and that any discrepancies from the study as planned have been explained.

**Contributors** AM, RP and MCB conceived and designed the analysis and AM is the guarantor of the overall content. RP led the patient and public involvement activities. LC, CG and MCB conducted the analysis. All authors drafted or gave substantive comments on the manuscript. The corresponding author attests that all listed authors meet authorship criteria and that no others meeting the criteria have been omitted. The authors are grateful to Miranda Scanlon and Dharmintra Pasupathy for comments on the manuscript and to the members of the public involvement and engagement group and our project advisory group for their guidance and input.

**Funding** The work described here was funded by the Economic and Social Research Council (ESRC) under its Secondary Data Analysis Initiative. Implications of time of day and day of the week for the outcome of birth (ES/S010785/1). It draws on earlier work 'Births and their outcomes by time, day and year: a retrospective birth cohort data linkage study', funded by the Health Services and Delivery Research Programme grant 12/136/93. The views expressed are those of the authors and not necessarily those of the ESRC.

**Competing interests** None declared.

**Patient and public involvement** Patients and/or the public were involved in the design, or conduct, or reporting, or dissemination plans of this research. Refer to the Methods section for further details.

**Patient consent for publication** Not applicable.

**Ethics approval** Ethics approval 05/Q0603/108 and subsequent substantial amendments were granted by East London and City Local Research Ethics Committee 1 and its successors. Use of patient identifiable data without consent Permission to use patient identifiable data without consent under Section 60 of the Health and Social Care Act 2001 was initially granted by the Patient Information Advisory Group PIAG 2-10(g)/2005. Renewals and amendments under Section 251 of the Health Service (Control of Patient Information) Regulations 2002 were granted by its successor bodies, the Ethics and Confidentiality Committee of the National Information Governance Board and the Confidentiality Advisory Group of the Health Research Authority. A second permission CAG 9-08(b)2014 to use patient identifiable data without consent under Section 251 of the Health Service (Control of Patient Information) Regulations 2002 to create a research database held at the Office for National Statistics for analyses relating to inequalities in the outcome of pregnancy and to inform maternity service users about the outcome of midwifery, obstetric and neonatal care was granted by the Confidentiality Advisory

Group of the Health Research Authority. Use of the Office for National Statistics Secure Research Service Permission to access data from the Office for National Statistics in the VML, now known as the Secure Research Service was granted by ONS's Microdata Release Panel, now superseded by its Research Advisory Panel. All members of the research team successfully applied for ONS Approved Researcher Status. The work is covered by the following disclaimer: 'This work was produced using statistical data from ONS. The use of the ONS statistical data in this work does not imply the endorsement of the ONS in relation to the interpretation or analysis of the statistical data. This work uses research datasets which may not exactly reproduce National Statistics aggregates.' Permission to use data from the Health and Social Care Information Centre Permission to link and analyse data held by the Health and Social Care Information Centre, subsequently known as NHS Digital and now part of NHS England, was granted under Data Sharing Agreements NIC-273840-N0N0N and subsequently under DARS-NIC-10094-P6P4B-v6.7 – Linkage, analysis and dissemination of national birth and maternity data for England and Wales.

**Provenance and peer review** Not commissioned; externally peer reviewed.

**Data availability statement** Data may be obtained from a third party and are not publicly available. The authors do not have permission to supply data or identifiable information to third parties, including other researchers, but they have permission under Regulation 5 of the Health Service (Control of Patient Information) Regulations 2002 to analyse confidential patient information for England and Wales without consent and create a research database which could be accessed by other researchers using the Secure Research Service at the Office for National Statistics. The information held includes the subset of data and the R computing code used to undertake the analyses described in this article. Anyone wishing to access the linked datasets for research purposes should apply to the Office for National Statistics and NHS England as well as to the Health Research Authority via the Confidentiality Advisory Group to access confidential patient information without consent. In the first instance, enquiries about access to the data should be addressed to the principal investigator, Alison Macfarlane (A.J.Macfarlane@city.ac. uk).

**ORCID iDs**
Christopher Grollman http://orcid.org/0000-0002-6950-1837
Alison Macfarlane http://orcid.org/0000-0003-0977-7214

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
