## [Reviewer comments · BMJ Open]

This paper was submitted to a another journal from BMJ but declined for publication following peer review. The authors addressed the reviewers' comments and submitted the revised paper to BMJ Open. The paper was subsequently accepted for publication at BMJ Open.

ARTICLE DETAILS

TITLE (PROVISIONAL)	Neonatal mortality in NHS maternity units by timing and mode of birth: a retrospective linked cohort study
AUTHORS	Carty, Lucy; Grollman, Christopher; Plachcinski, Rachel; Cortina Borja, Mario; Macfarlane, Alison

VERSION 1 – REVIEW

REVIEWER	de Jonge, Ank AVAG, APH research institute, Amsterdam UMC, locatie VUMC, Midwifery Science
REVIEW RETURNED	11-Sep-2022

GENERAL COMMENTS	Thank you for asking me to review this very interesting paper. The use of a very large dataset enables the study of small differences in mortality rates which is very useful. I have some comments that may be helpful in improving the paper. When I mention page numbers I mean the number given in the manuscript at the top of the page (page... of 37). Abstract P3, l32 I do not understand where the conclusion comes from that obstetric staff should be planned in such a way that unusual emergencies can be dealt with (grammar needs adjusting too). The authors did not examine why they found a higher neonatal mortality rate among women with an emergency CS without labour. As they discuss in the paper, it might be, for example, that patients' phone late – in that case, staffing is not the problem. Please adjust the abstract after addressing the comments about the main text. Introduction P.4, L.6: Please change 'perinatal rates higher' into 'higher perinatal rates'. P4, L.20 I would say mode of birth instead of method of birth as spontaneous vaginal birth is not a method. P4, L19/20 The objective of the study is focused on neonatal mortality, but infant deaths up to one year are also shown. Please decide whether you want to look at infant deaths (in that case mention this in the objective as well) or at neonatal deaths only. Considering the focus is on intrapartum events, it makes most sense to limit the analyses to neonatal mortality up to 28 days. This will also reduce the number of results and therefore it will be easier to combine tables (see further).
--

Methods

P4, L. 49: derivation and linkage bias analysis was reported elsewhere. Please give a brief summary here of the established quality (or not) of the linked data.

P4, L51 Please explain here why you did not include intrapartum deaths.

P.5, L.12 '... likely to be effected', please remove 'to'

P.5, L.30 and further. Please give a rationale for the definition of week- and weekend days. Is this based on the off duty schedule of healthcare workers? It seems strange to define time up to Saturday morning 7 am as a weeknight. I would recommend a sensitivity analysis to see if results are similar if this night is added to the group of births in the weekend.

P.6, L.24: 15% births should be 15% of births

P.6, L.25: and included them – 'we' is missing

P.5 and 6: the involvement of the public in the research is impressive! This paragraph can be shortened by focusing on public involvement for this particular study mainly.

Results

It would be very helpful to include a flow diagram to show how the study population was selected. Table 1 is very confusing as it includes deaths that are not included in the main analyses. I think table 2 should be table 1 and only this study group should be defined as the study population. For example, in the methods you describe that deaths due to congenital abnormalities were removed, but table 1 shows the number of deaths in this group, which is very confusing.

P.7, l.22 that were missing – please change were into had

P.7, l.35 from 'this is likely....' – this should be in the discussion, not in the results section

Table 5, 6 and 7 can be merged as there is a lot of overlap. I would advise to add 2 columns in table 5 with adjusted rate ratio's for all modes of birth after 1) adjusting for characteristics 2) adjusting for obstetric factors as well. Also, please show 2 decimals for all odds ratio's and be consistent in the figures shown. There are small differences in results in the three tables that should display the same findings.

P.12, l.49 Please add the comment that these data were not shown.

P.13, l.6 You cannot say that an emergency CS causes a neonatal death. You have only shown associations, not causation as you used registration data.

Discussion

Please restructure the discussion:

- Give the main results of the study only without discussing them in view of the literature

- Add a paragraph in which you write the strengths and limitations

- Add a paragraph in which you interpret the results in view of the literature – describe the recommendations that follow for practice and further research; these recommendations may be in a separate paragraph but may also follow logically from the discussion of the results. As it is now, the recommendations paragraph starts with a recommendation followed by a discussion of the findings. This is confusing. For example, it is not clear where the recommendation at p17, l7 and further comes from (see earlier comment about abstract). Particularly as you state in l.49 at the same page that further research is needed as the reasons for emergency CS without labour being associated with higher mortality rates are unclear.

	P.14, L50 The explanation about why elective CS were not included in certain analyses should be moved to the methods or results section. As the authors state, a planned CS outside working hours is usually not really planned but is either an emergency CS which was coded wrongly or was a CS in a woman who had a planned CS but went into labour before the operation took place. I suspect, many of the emergency CS belong to the same category – the authors explain that most inconsistencies between different datasets were due to one source recoding an emergency CS and the other a planned CS. Therefore, I would merge the categories planned and emergency CS outside working hours. P.15, l.5/6 Please correct the grammar of the sentence. P.15, l.31 Odds ratio should be plural.
--	---

REVIEWER	Bion, Julian University of Birmingham, Intensive Care Medicine
REVIEW RETURNED	10-Mar-2023

GENERAL COMMENTS	The rationale for this study is the ‘weekend effect’, the excess mortality amongst adult patients admitted to hospital at weekends, or undergoing surgery at weekends. The ‘cause’ of the weekend effect in adults has been substantially explicated by the HiSLAC study group, who have shown that patients admitted at weekends are sicker, frailer, more likely to receive palliative care, and much less likely to have been referred by a general practitioners in the community than patients admitted on weekdays [https://qualitysafety.bmj.com/content/30/7/536]. They do not receive poorer in-hospital care at weekends. This points to a community cause, and the differences become more marked with the passage of time. It would therefore be of interest first, to determine whether the ‘weekend effect’ exists for childbirth, and second whether this has its origins in hospital or community care. The authors have examined neonatal mortality rates in relation to the timing of birth in 6,054,536 singleton births in NHS maternity units in England between 2005 and 2014. Data linkage was performed using birth registration and birth notification from the Office of National Statistics, linked to maternity hospital episode statistics (HES). The authors have grouped together births occurring on weekdays 7am-7pm, and compared these with births occurring during ‘non-working hours’ (all other times - weekday nights, weekends, bank holidays). Neonatal deaths were those occurring within 28 days of a live birth. They have further disaggregated births at night from those during the day, for weekdays and weekends. Mortality was adjusted for a range of known risk factors. Other than operative intervention, the data set did not permit examination of processes of care, but they were able to examine secular changes from 2005 to 2014. Their main finding is that while there is no excess mortality associated with births ‘out-of-hours’ there is an excess mortality amongst neonates born at night by emergency caesarean section, and this is attributable to those who presented without prior labour. The excess mortality risk was 56% for weekday nights and 75% for weekends including nights and bank holidays. This is equivalent to an additional 46 deaths over the ten year period. This subpopulation consists of mothers and babies in whom an emergency has arisen presumably without much prior warning, or
---

perhaps those who have not sought pre-natal care but are higher risk. It would be helpful if the authors could describe (very briefly) the possible characteristics or pathways involved for this group, perhaps expanding the two sentences in the discussion section at the end of the paragraph on 'Meaning of the study...' where the authors speculate on possible causes for higher mortality. They consider community-based factors only very briefly. They may like to consider the article by Bion et al which identifies a community contribution towards the weekend effect in adults.

Was failure to record time of birth (and thereby excluded from the analysis) evenly distributed between 'in-hours' and 'out-of-hours'?

By adjusting for the former Strategic Health Authority Region of England where the birth took place, might you have adjusted away important factors in the community which could have contributed to adverse outcomes, such as deprivation or poorer community services?

Similarly, it appears that you did not factor in (or out) post code. Was there a reason for not doing so? It is contained in HES.

Minor points:

I don't much like the term 'non-working' hours. It is a misnomer. 'Out of hours' is a more conventional short-hand fudge.

Methods appear in Results – see section on "Absolute risks from emergency caesarean without labour during non-working hours", page 14.

Results appear in the Discussion section: see para 2 of the Discussion on page 14.

Strengths and Weaknesses: The first paragraph does not directly address this issue: it compares your research with the Scottish study. The subsequent paragraphs in this section are quite 'narrative' in presentation. I would find it easier to extract the main messages if you included a subheading for each paragraph indicating how your study is an improvement on others (for example: 'Data on stillbirths' preceding the second long paragraph on page 16). This whole section is also very long, and would benefit from greater concision. Strengths are mixed up with weaknesses.

VERSION 1 – AUTHOR RESPONSE

No.	Section	Reviewer's comment	Response
1	Overall	R2: I don't much like the term 'non-working' hours. It is a misnomer. 'Out of hours' is a more conventional short-hand fudge.	We have made this change.
2	Overall	Eds: - Please remove the summary box from your paper - this is not required for submission to BMJ Open.	Done
3	Overall	Eds: - Please include a 'Strengths and limitations' section (after the abstract). This section should contain five short bullet points, no longer than one sentence each, that relate specifically to the methods. The results of the study should not be summarised here.	Added
4	Abstract	R1: P3, l32 I do not understand where the conclusion comes from that obstetric staff should be planned in such a way that unusual emergencies can be dealt with (grammar needs adjusting too). The authors did not examine why they found a higher neonatal mortality rate among women with an emergency CS without labour. As they discuss in the paper, it might be, for example, that patients' phone late – in that case, staffing is not the problem.	We have rephrased this sentence to read "Further research should investigate the potential contribution of care-seeking and community-based factors as well as the adequacy of staffing for managing these relatively unusual emergencies."
5	Abstract	R1: Please adjust the abstract after addressing the comments about the main text.	We have done so, thank you.
6	Introduction	R1: P.4, L.6: Please change 'perinatal rates higher' into 'higher perinatal rates'.	Done
7	Introduction	R1: P4, L.20 I would say mode of birth instead of method of birth as spontaneous vaginal birth is not a method.	Changed throughout

8	Introduction	R1: P4, L19/20 The objective of the study is focused on neonatal mortality, but infant deaths up to one year are also shown. Please decide whether you want to look at infant deaths (in that case mention this in the objective as well) or at neonatal deaths only. Considering the focus is on intrapartum events, it makes most sense to limit the analyses to neonatal mortality up to 28 days. This will also reduce the number of results and therefore it will be easier to combine tables (see further). We have removed the infant and postneonatal deaths from that table.
9	Methods	R1: P4, L. 49: derivation and linkage bias analysis was reported elsewhere. Please give a brief summary here of the established quality (or not) of the linked data. We have added a statement "In summary, it was possible to link over 94 per cent of birth registration linked to notification records to HES delivery and birth records. The linkage rate increased over time as the quality of Maternity. of Maternity HES improved."
10	Methods	R1: P4, L51 Please explain here why you did not include intrapartum deaths. Added the following text: "We chose not to include intrapartum stillbirths in our outcome measure as the timing of stillbirth is poorly recorded in many cases; there are as many stillbirths of indeterminate timing as there are stillbirths recorded as intrapartum. By contrast, virtually all neonatal deaths are unambiguous and we have higher confidence in the completeness of that population."
11	Methods	R1: P.5, L.12 '... likely to be effected', please remove 'to' Revised from "as this category of death would be most likely to be affected " to "as this is the category of death most likely to be affected"

12	Methods	R1: P.5, L.30 and further. Please give a rationale for the definition of week- and weekend days. Is this based on the off duty schedule of healthcare workers? It seems strange to define time up to Saturday morning 7 am as a weeknight. I would recommend a sensitivity analysis to see if results are similar if this night is added to the group of births in the weekend.	Apologies, there was an error in the manuscript: Saturday before 7am is not classified as a weeknight. (It had been in an earlier iteration of the timing variable.) We have changed that sentence to read: "In breaking down non-working hours further, the hours from midnight to 7am on a Monday were classified as a weekend/holiday night to reflect the fact that people giving birth in that period would have had access to weekend staffing only in the preceding 48 hours."
13	Methods	R1: P.6, L.24: 15% births should be 15% of births P.6, L.25: and included them – 'we' is missing	Done.
14	Methods	R2: P.5 and 6: the involvement of the public in the research is impressive! This paragraph can be shortened by focusing on public involvement for this particular study mainly.	Thank you - we have substantially shortened this section (from 497 to 244 words).
15	Methods	R2: By adjusting for the former Strategic Health Authority Region of England where the birth took place, might you have adjusted away important factors in the community which could have contributed to adverse outcomes, such as deprivation or poorer community services?	Thank you for this suggestion, which prompted us to conduct various sensitivity analyses. The results for the timing variable are unchanged in all of these scenarios, which involved running the main model: with and without trust (specifying GLM when not using the random effects term with trust); with and without SHA region; with an interaction term for region*timing and with an interaction term for trust*timing. We have added a comment in the end of the discussion to say "There may be scope to use the City Birth Cohort for more fine-grained investigation including of variation by trust, if power could be increased using a neonatal near-miss outcome rather than mortality."
16	Methods	R2: Similarly, it appears that you did not factor in (or out) post code. Was there a reason for not doing so? It is contained in HES.	We have added a note about IMD, which we did investigate but was not associated with the outcome and didn't affect the point estimate for the timing variable.

17	Results	R1: It would be very helpful to include a flow diagram to show how the study population was selected. Table 1 is very confusing as it includes deaths that are not included in the main analyses. I think table 2 should be table 1 and only this study group should be defined as the study population. For example, in the methods you describe that deaths due to congenital abnormalities were removed, but table 1 shows the number of deaths in this group, which is very confusing.	We appreciate the critical view on this and hope our revision makes it clearer. We think it is valuable to place the 'anoxia, asphyxia and trauma' deaths in the context of wider cause-specific neonatal mortality, but have changed the order in which we present the various population exclusions in order to make it read more straightforwardly. Please let us know if this makes it adequately clearer!
18	Results	R1: P.7, I.22 that were missing – please change were into had	Done.
19	Results	R1: P.7, I.35 from 'this is likely....' – this should be in the discussion, not in the results section	Change made - thank you.
20	Results	R1: Table 5, 6 and 7 can be merged as there is a lot of overlap. I would advise to add 2 columns in table 5 with adjusted rate ratio's for all modes of birth after 1) adjusting for characteristics 2) adjusting for obstetric factors as well. Also, please show 2 decimals for all odds ratio's and be consistent in the figures shown. There are small differences in results in the three tables that should display the same findings.	Thank you - we have merged tables 6 and 7, presenting modelled odds ratios. We consider it appropriate to keep these separate from table 5 presenting descriptive rate ratios but are willing to find a way to amalgamate these if you feel strongly!
21	Results	R1: P.12, I.49 Please add the comment that these data were not shown.	Done.
22	Results	R2: P.13, I.6 You cannot say that an emergency CS causes a neonatal death. You have only shown associations, not causation as you used registration data.	Changed to "be associated with". Also in the discussion changed "we estimate that 46 excess deaths are likely to be attributable to being outside of working hours across the ten years" to "we estimate that 46 excess deaths are likely to be associated with being born outside of working hours across the ten years".

23	Results	R2: Was failure to record time of birth (and thereby excluded from the analysis) evenly distributed between 'in-hours' and 'out-of-hours'?	We cannot know whether these births were in-hours or out-of-hours as that categorisation depends on knowing the time of birth. We do know that this failure to record was not associated with the outcome (p 8/37, line 24: "We did not find any evidence of association between missing time of birth and neonatal mortality (Chi squared p = 0.75).").
24	Results	R2: Methods appear in Results – see section on “Absolute risks from emergency caesarean without labour during non-working hours”, page 14.	Added methods to Methods and removed from Results
25	Discussion	R1: Please restructure the discussion:	We have done so, thank you.
26	Discussion	R1: - Give the main results of the study only without discussing them in view of the literature	We have made the structure of the discussion more ordered and clear.
27	Discussion	R1: - Add a paragraph in which you write the strengths and limitations	We have added a more structured discussion of strengths and limitations in the discussion, as well as the methods strengths and limitations asked for by the editors at the beginning of the paper.
28	Discussion	R1: - Add a paragraph in which you interpret the results in view of the literature – describe the recommendations that follow for practice and further research; these recommendations may be in a separate paragraph but may also follow logically from the discussion of the results. As it is now, the recommendations paragraph starts with a recommendation followed by a discussion of the findings. This is confusing. For example, it is not clear where the recommendation at p17, 17 and further comes from (see earlier comment about abstract). Particularly as you state in I.49 at the same page that further research is needed as the reasons for emergency CS without labour being associated with higher mortality rates are unclear.	Thank you - we have added subheading in our strengths and limitations section to make the document read more smoothly. We have removed the specific comment about 'helplines'.

				Thank you for raising this. Your comment prompted us to conduct a sensitivity analysis on this question. Under 3% of elective Cs are recorded as occurring at night. If we categorise these as emergency Cs they constitute 2% of all emergency Cs.
		R1: P.14, L50	The explanation about why elective CS were not included in certain analyses should be moved to the methods or results section. As the authors state, a planned CS outside working hours is usually not really planned but is either an emergency CS which was coded wrongly or was a CS in a woman who had a planned CS but went into labour before the operation took place. I suspect, many of the emergency CS belong to the same category – the authors explain that most inconsistencies between different datasets were due to one source recoding an emergency CS and the other a planned CS. Therefore, I would merge the categories planned and emergency CS outside working hours.	It is possible that these cases were all truly emergency Cs, and indeed some have spontaneous or induced onset recorded which supports that view. It is also possible that some cases with onset recorded as caesarean, or missing that data, are miscategorised as having happened at night.
29	Discussion			If we include all elective C at night as emergencies, it doesn't affect the overall interpretation. It does reduce the odds ratios slightly for the two categories of birth at night.
30	Discussion	R1: P.15, l.5/6	Please correct the grammar of the sentence.	Would you like us to include the sensitivity analysis in the supplementary materials? Corrected
31	Discussion	R1: P.15, l.31	Odds ratio should be plural.	Corrected
		R2:	This subpopulation consists of mothers and babies in whom an emergency has arisen presumably without much prior warning, or perhaps those who have not sought pre-natal care but are higher risk. It would be helpful if the authors could describe (very briefly) the possible characteristics or pathways involved for this group, perhaps expanding the two sentences in the discussion section at the end of the paragraph on 'Meaning of the study...' where the authors speculate on possible causes for higher mortality. They consider community-based factors only very briefly. They may like to consider the article by Bion et al which identifies a community contribution towards the weekend effect in adults.	As we have now acknowledged in the new 'Strengths and limitations' section, we were unable to consider community factors because our analysis used hospital based administrative data. This contrasts with the analysis by Bion et al which was a case note review and thus could consider a much wider range of information, including factors outside hospital, but derived from a much smaller dataset. We have expanded our acknowledgement of this.
32	Discussion			

33	Discussion	R2: Results appear in the Discussion section: see para 2 of the Discussion on page 14.	Thank you - added the 2.2% figure into the results section.
34	Discussion	R2: Strengths and Weaknesses: The first paragraph does not directly address this issue: it compares your research with the Scottish study. The subsequent paragraphs in this section are quite 'narrative' in presentation. I would find it easier to extract the main messages if you included a subheading for each paragraph indicating how your study is an improvement on others (for example: 'Data on stillbirths' preceding the second long paragraph on page 16). This whole section is also very long, and would benefit from greater concision. Strengths are mixed up with weaknesses.	We have substantially restructured the discussion and included subheadings, and shortened in places.